# A Unifying Perspective on Neighbor Embeddings along the Attraction-Repulsion Spectrum

## Abstract

Neighbor embeddings are a family of methods for visualizing complex high-dimensional datasets using kNN graphs. To find the low-dimensional embedding, these algorithms combine an attractive force between neighboring pairs of points with a repulsive force between all points. One of the most popular examples of such algorithms is t-SNE. Here we empirically show that changing the balance between the attractive and the repulsive forces in t-SNE yields a spectrum of embeddings, which is characterized by a simple trade-off: stronger attraction can better represent continuous manifold structures, while stronger repulsion can better represent discrete cluster structures. We find that UMAP embeddings correspond to t-SNE with increased attraction; mathematical analysis shows that this is because the negative sampling optimisation strategy employed by UMAP strongly lowers the effective repulsion. Likewise, ForceAtlas2, commonly used for visualizing developmental single-cell transcriptomic data, yields embeddings corresponding to t-SNE with the attraction increased even more. At the extreme of this spectrum lies Laplacian Eigenmaps, corresponding to zero repulsion. Our results demonstrate that many prominent neighbor embedding algorithms can be placed onto this attraction-repulsion spectrum, and highlight the inherent trade-offs between them.

## 1 Introduction

T-distributed stochastic neighbor embedding (t-SNE) (van der Maaten & Hinton, 2008) is arguably among the most popular methods for low-dimensional visualizations of complex high-dimensional datasets. It defines pairwise similarities called *affinities* between points in the high-dimensional space and aims to arrange the points in a low-dimensional space to match these affinities (Hinton & Roweis, 2003). Affinities decay exponentially with high-dimensional distance, making them infinitesimal for most pairs of points and making the $n \times n$ affinity matrix effectively sparse. Efficient implementations of t-SNE suitable for large sample sizes $n$ (van der Maaten, 2014; Linderman et al., 2019) explicitly truncate the affinities and use the $k$-nearest-neighbor (kNN) graph of the data with $k \ll n$ as the input.

We use the term *neighbor embedding* (NE) to refer to all dimensionality reduction methods that operate on the kNN graph of the data and aim to preserve neighborhood relationships (Yang et al., 2013; 2014). A prominent recent example of this class of algorithms is UMAP (McInnes et al., 2018), which has become popular in applied fields such as single-cell transcriptomics (Becht et al., 2019). It is based on stochastic optimization and typically produces more compact clusters than t-SNE.

Another example of neighbor embeddings are force-directed graph layouts (Noack, 2007; 2009), originally developed for graph drawing. One specific algorithm called ForceAtlas2 (Jacomy et al., 2014) has recently gained popularity in the single-cell transcriptomic community to visualize datasets capturing cells at different stages of development (Weinreb et al., 2018; 2020; Wagner et al., 2018a; Tusi et al., 2018; Kanton et al., 2019; Sharma et al., 2020).

Here we provide a unifying account of these algorithms. We studied the spectrum of t-SNE embeddings that are obtained when increasing/decreasing the attractive forces between kNN graph neighbors, thereby changing the balance between attraction and repulsion. This led to a trade-off between faithful representations of continuous and discrete structures (Figure 1). Remarkably, we found that ForceAtlas2 and UMAP could both be accurately positioned on this spectrum (Figure 1). For UMAP, we used mathematical analysis and Barnes-Hut re-implementation to show that increased attraction is due to the negative sampling optimisation strategy.

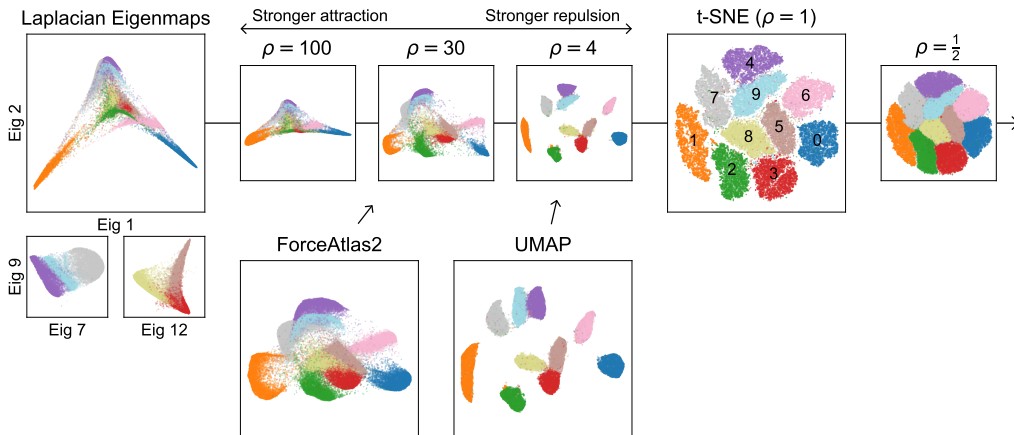

Figure 1: **Attraction-repulsion spectrum for the MNIST data.** Different embeddings of the full MNIST dataset of hand-written digits ($n = 70\,000$); colors correspond to the digit value as shown in the t-SNE panel. Multiplying all attractive forces by an exaggeration factor $\rho$ yields a spectrum of embeddings. Values below 1 yield inflated clusters. Small values above 1 yield more compact clusters. Higher values make multiple clusters merge, with $\rho \to \infty$ corresponding to Laplacian Eigenmaps. Insets show two subsets of digits separated in higher Laplacian eigenvectors. UMAP is very similar to $\rho \approx 4$. ForceAtlas2 is very similar to $\rho \approx 30$.

## 2 RELATED WORK

Various trade-offs in t-SNE generalizations have been studied previously (Yang et al., 2009; Kobak et al., 2020; Venna et al., 2010; Amid et al., 2015; Amid & Warmuth, 2019; Narayan et al., 2015; Im et al., 2018), but our work is the first to study the *exaggeration*-induced trade-off. Prior work used 'early exaggeration' only as an optimisation trick (van der Maaten & Hinton, 2008) that allows to separate well-defined clusters (Linderman & Steinerberger, 2019; Arora et al., 2018).

Carreira-Perpiñán (2010) introduced *elastic embedding* algorithm that has an explicit parameter $\lambda$ controlling the attraction-repulsion balance. However, that paper suggests slowly increasing $\lambda$ during optimization, as an optimisation trick similar to the early exaggeration, and does not discuss tradeoffs between high and low values of $\lambda$.

Our results on UMAP go against the common wisdom on what makes UMAP perform as it does (McInnes et al., 2018; Becht et al., 2019). No previous work suggested that negative sampling may have a drastic effect on the resulting embedding.

## 3 NEIGHBOR EMBEDDINGS

We first cast t-SNE, UMAP, and ForceAtlas2 in a common mathematical framework, using consistent notation and highlighting the similarities between the algorithms, before we investigate the relationships between them empirically and analytically in more detail. We denote the original high-dimensional points as $\mathbf{x}_i$ and their low-dimensional positions as $\mathbf{y}_i$.

### 3.1 T-SNE

T-SNE measures similarities between points by *affinities* $v_{ij}$ and *normalized affinities* $p_{ij}$:

$$p_{ij} = \frac{v_{ij}}{n}, \quad v_{ij} = \frac{p_{i|j} + p_{j|i}}{2}, \quad p_{j|i} = \frac{v_{j|i}}{\sum_{k \neq i} v_{k|i}}, \quad v_{j|i} = \exp\left(-\frac{\|\mathbf{x}_i - \mathbf{x}_j\|^2}{2\sigma_i^2}\right). \quad (1)$$

For fixed $i$, $p_{j|i}$ is a probability distribution over all points $j \neq i$ (all $p_{i|i}$ are set to zero), and the variance of the Gaussian kernel $\sigma_i^2$ is chosen to yield a pre-specified value of the perplexity

of this probability distribution, $\mathcal{P} = 2^{\mathcal{H}}$, where $\mathcal{H} = -\sum_{j \neq i} p_{j|i} \log_2 p_{j|i}$. The affinities $v_{ij}$ are normalized by $n$ for $p_{ij}$ to form a probability distribution on the set of all pairs of points $(i, j)$. Modern implementations (van der Maaten, 2014; Linderman et al., 2019) construct a kNN graph with $k = 3\mathcal{P}$ neighbors and only consider affinities between connected nodes as non-zero. The default perplexity value in most implementations is $\mathcal{P} = 30$.

Similarities in the low-dimensional space are defined as

$$q_{ij} = \frac{w_{ij}}{Z}, \quad w_{ij} = \frac{1}{1 + d_{ij}^2}, \quad d_{ij} = \|\mathbf{y}_i - \mathbf{y}_j\|, \quad Z = \sum_{k \neq l} w_{kl}, \tag{2}$$

with all $q_{ii}$ set to 0. The points $\mathbf{y}_i$ are then rearranged in order to minimise the Kullback-Leibler (KL) divergence $\mathcal{D}_{\text{KL}}(\{p_{ij}\} \| \{q_{ij}\}) = \sum_{i,j} p_{ij} \log(p_{ij}/q_{ij})$ between $p_{ij}$ and $q_{ij}$:

$$\mathcal{L}_{\text{t-SNE}} = -\sum_{i,j} p_{ij} \log \frac{w_{ij}}{Z} = -\sum_{i,j} p_{ij} \log w_{ij} + \log \sum_{i,j} w_{ij}, \tag{3}$$

where we dropped constant terms and took into account that $\sum p_{ij} = 1$. The first term can be interpreted as contributing attractive forces to the gradient while the second term yields repulsive forces. Using $\partial w_{ij}/\partial \mathbf{y}_i = -2w_{ij}^2(\mathbf{y}_i - \mathbf{y}_j)$, the gradient, up to a constant factor, can be written as:

$$\frac{\partial \mathcal{L}_{\text{t-SNE}}}{\partial \mathbf{y}_i} \sim \sum_j v_{ij} w_{ij}(\mathbf{y}_i - \mathbf{y}_j) - \frac{n}{Z} \sum_j w_{ij}^2(\mathbf{y}_i - \mathbf{y}_j). \tag{4}$$

### 3.2 Exaggeration in t-SNE

A standard optimisation trick for t-SNE called *early exaggeration* (van der Maaten & Hinton, 2008; van der Maaten, 2014) is to multiply the first sum in the gradient by a factor $\rho = 12$ during the initial iterations of gradient descent. This increases the attractive forces and allows similar points to gather into clusters more effectively. Carreira-Perpiñán (2010) and Linderman & Steinerberger (2019) noticed that the attractive term in the t-SNE loss function is related to the loss function of Laplacian eigenmaps (LE) (Belkin & Niyogi, 2002; Coifman & Lafon, 2006). Indeed, if $\rho \to \infty$, the relative repulsion strength goes to zero and the embedding shrinks to a point with all $w_{ij} \to 1$. This implies that, asymptotically, gradient descent becomes equivalent to Markov chain iterations with the transition matrix closely related to the graph Laplacian $\mathbf{L} = \mathbf{D} - \mathbf{V}$ of the affinity matrix $\mathbf{V} = [v_{ij}]$ (here $\mathbf{D}$ is diagonal matrix with row sums of $\mathbf{V}$; see Appendix). The entire embedding shrinks to a single point, but the leading eigenvectors of the Laplacian shrink the slowest. This makes t-SNE with $\rho \to \infty$ produce embeddings very similar to LE, which computes the leading eigenvectors of the normalized Laplacian (see Appendix and Figure 1).

This theoretical finding immediately suggests that it might be interesting to study t-SNE with exaggeration $\rho > 1$ not only as an optimisation trick, but in itself, as an intermediate method between LE and standard t-SNE. The gradient of t-SNE with exaggeration can be written as

$$\frac{\partial \mathcal{L}_{\text{t-SNE}}(\rho)}{\partial \mathbf{y}_i} \sim \sum_j v_{ij} w_{ij}(\mathbf{y}_i - \mathbf{y}_j) - \frac{n}{\rho Z} \sum_j w_{ij}^2(\mathbf{y}_i - \mathbf{y}_j) \tag{5}$$

and the corresponding loss function is

$$\mathcal{L}_{\text{t-SNE}}(\rho) = \mathcal{D}_{\text{KL}}\left(\{p_{ij}\} \| \{w_{ij}/Z^{\frac{1}{\rho}}\}\right) = \sum_{i,j} p_{ij} \log \frac{p_{ij}}{w_{ij}/Z^{\frac{1}{\rho}}}. \tag{6}$$

### 3.3 UMAP

Using the same notation as above, UMAP optimizes the cross-entropy loss between $v_{ij}$ and $w_{ij}$, without normalizing them into probabilities:

$$\mathcal{L}_{\text{UMAP}} = \sum_{i,j} \left[ v_{ij} \log \frac{v_{ij}}{w_{ij}} + (1 - v_{ij}) \log \frac{1 - v_{ij}}{1 - w_{ij}} \right], \tag{7}$$

where the $1 - v_{ij}$ term is approximated by 1 as most $v_{ij}$ are 0. Note that UMAP differs from t-SNE in how exactly it defines $v_{ij}$ but this difference is negligible, at least for the data considered here.[1]

Dropping constant terms, we obtain

$$\mathcal{L}_{\text{UMAP}} \sim -\sum_{i,j} v_{ij} \log w_{ij} - \sum_{i,j} \log(1 - w_{ij}), \tag{8}$$

which is the same loss function as the one introduced earlier by LargeVis (Tang et al., 2016). The first term, corresponding to attractive forces, is the same as in t-SNE, but the second, repulsive, term is different. Taking $w_{ij} = 1/(1 + d_{ij}^2)$ as in t-SNE,[2] the UMAP gradient is given by

$$\frac{\partial \mathcal{L}_{\text{UMAP}}}{\partial \mathbf{y}_i} \sim \sum_j v_{ij} w_{ij}(\mathbf{y}_i - \mathbf{y}_j) - \sum_j \frac{1}{d_{ij}^2 + \epsilon} w_{ij}(\mathbf{y}_i - \mathbf{y}_j), \tag{9}$$

where $\epsilon = 0.001$ is added to the denominator to prevent numerical problems for $d_{ij} \approx 0$. If $\epsilon = 1$ (which does not strongly affect the result; Figure S1), the gradient becomes identical to the t-SNE gradient, up to the $n/Z$ factor in front of the repulsive forces. Moreover, UMAP allows to use an arbitrary $\gamma$ factor in front of the repulsive forces, which makes it easier to compare the loss functions[3]:

$$\frac{\partial \mathcal{L}_{\text{UMAP}}(\gamma)}{\partial \mathbf{y}_i} \sim \sum_j v_{ij} w_{ij}(\mathbf{y}_i - \mathbf{y}_j) - \gamma \sum_j \frac{1}{d_{ij}^2 + \epsilon} w_{ij}(\mathbf{y}_i - \mathbf{y}_j). \tag{10}$$

Whereas it is possible to approximate the full repulsive term with the same techniques as used in t-SNE (van der Maaten, 2014; Linderman et al., 2019), UMAP took a different approach and followed LargeVis in using *negative sampling* (Mikolov et al., 2013) of repulsive forces: on each gradient descent iteration, only a small number $\nu$ of randomly picked repulsive forces are applied to each point for each of the $\sim k$ attractive forces that it feels. Other repulsive terms are ignored. The default value is $\nu = 5$. The effect of this negative sampling on the resulting embedding has not been studied before.

### 3.4 FORCEATLAS2

Force-directed graph layouts are usually introduced directly via attractive and repulsive forces, even though it is easy to write down a suitable loss function (Noack, 2007). ForceAtlas2 (FA2) has attractive forces proportional to $d_{ij}$ and repulsive forces proportional to $1/d_{ij}$ (Jacomy et al., 2014):

$$\frac{\partial \mathcal{L}_{\text{FA2}}}{\partial \mathbf{y}_i} = \sum_j v_{ij}(\mathbf{y}_i - \mathbf{y}_j) - \sum_j \frac{(h_i + 1)(h_j + 1)}{d_{ij}^2}(\mathbf{y}_i - \mathbf{y}_j), \tag{11}$$

where $h_i$ denotes the degree of node $i$ in the input graph. This is known as *edge repulsion* in the graph layout literature (Noack, 2007; 2009) and is important for embedding graphs that have nodes of very different degrees. For symmetrized kNN graphs, $h_i \approx k$, so $(h_i + 1)(h_j + 1)$ term contributes a roughly constant factor of $\sim k^2$ to the repulsive forces.

### 3.5 IMPLEMENTATION

All experiments were performed in Python. We ran all packages with default parameters, unless specified. We used openTSNE 0.4.4 (Policar et al., 2019), a Python reimplementation of FIt-SNE (Linderman et al., 2019). When using $\rho < 12$, we used the default early exaggeration with $\rho_{\text{early}} = 12$, and exaggeration $\rho$ for all subsequent iterations. For $\rho \geq 12$ no early exaggeration was used and exaggeration $\rho$ was applied throughout. The learning rate was set to $\eta = n/\max(\rho, \rho_{\text{early}})$ (Belkina et al., 2019). We used UMAP 0.4.0 with a Cauchy similarity kernel (i.e. setting a=b=1). For

---

[1]T-SNE uses an adaptive Gaussian kernel with default $\mathcal{P} = 30$ and $k = 90$. UMAP uses an adaptive Laplacian kernel with default $k = 15$. We found that for both algorithms one can use binary affinities with $k = 15$ without noticeably changing the outcome. Specifically, we used the kNN ($k = 15$) adjacency matrix $\mathbf{A} = [a_{ij}]$ to construct symmetric binary affinities $v_{ij} = (a_{ij} \lor a_{ji})/k$. When using these values, the t-SNE and UMAP embeddings stayed almost the same (Figure S1).

[2]UMAP uses $w_{ij} = 1/(1 + ad_{ij}^{2b})$ as an output kernel with $a \approx 1.6$ and $b \approx 0.9$ by default. Setting $a = b = 1$ does not qualitatively affect the result (Figure S1).

[3]LargeVis used $\gamma = 7$ but UMAP sets $\gamma = 1$ by default, as follows from its cross-entropy loss.

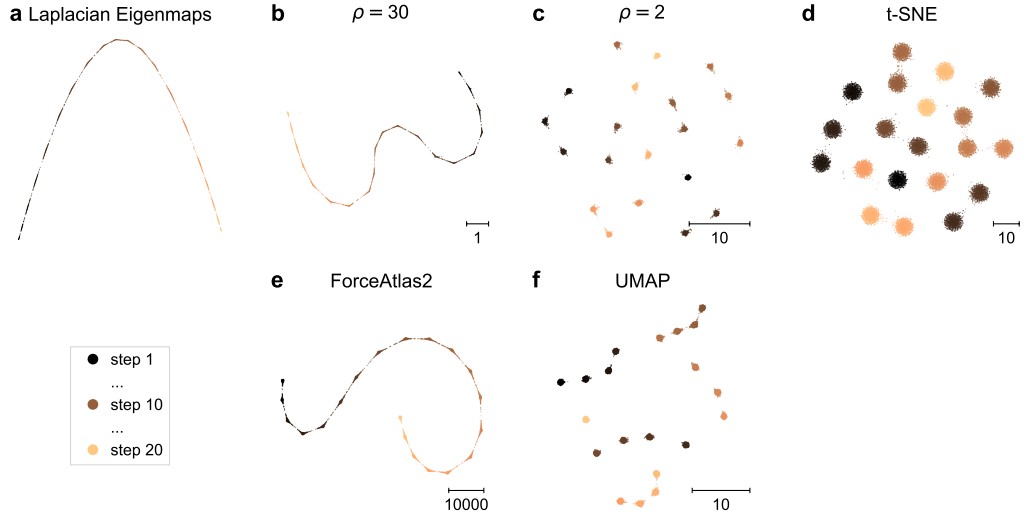

Figure 2: **Simulated data emulating a developmental trajectory.** The points were sampled from 20 isotropic 50-dimensional Gaussians, equally spaced along one axis such that only few inter-cluster edges exist in the kNN graph. Panels (b–f) used a shared random initialization.

FA2, we used `fa2` package (Chippada, 2017) that employs a Barnes-Hut approximation to speed up computation of the repulsive forces. The input to FA2 was the unweighted symmetrized approximate kNN graph $\mathbf{A} \vee \mathbf{A}^\top$, where $\mathbf{A}$ is the kNN adjacency matrix constructed with `Annoy` (Bernhardsson, 2013) with $k = 15$. All algorithms were optimized for 750 iterations, unless stated otherwise.

Unless stated otherwise, we used principal component analysis (PCA) initialisation to remove any differences due to initialization strategies (Kobak & Linderman, 2019) and to make all embeddings of the same dataset aligned to each other (Kobak & Berens, 2019). PCs 1/2 were scaled to standard deviation 0.0001 for t-SNE and to span 10 for UMAP to match the initialization scalings used in the respective implementations, and to a standard deviation of $10\,000$ for ForceAtlas2 to approximately match its final scale. Figure 2 uses random initialization. LE was computed using the `scikit-learn` (Pedregosa et al., 2011) implementation (`SpectralEmbedding`). The input was the same as the input to FA2. No initialisation was needed for LE. We flipped the signs of LE eigenvectors to orient them similarly to other embeddings, whenever necessary.

## 4 THE ATTRACTION-REPULSION SPECTRUM

We first investigated the relationships between the NE algorithms using the MNIST dataset of hand-written digits (sample size $n = 70\,000$; dimensionality $28 \times 28 = 784$, reduced to 50 with PCA; Figure 1). T-SNE produced an embedding where all ten digits were clearly separated into clusters with little white space between them, making it difficult to assess relationships between digits. Increasing attraction to $\rho = 4$ shrank the clusters and strongly increased the amount of white space; it also identified two groups of graphically similar digits: "4/7/9" and "3/5/8". Further increasing the attraction to $\rho = 30$ made all clusters connect together via some poorly written digits: e.g. cluster "6" connects to "5" and to "0". Even higher exaggeration made the embedding similar to Laplacian eigenmaps, in agreement with the theoretical prediction discussed above (Linderman et al., 2019). Here similar digit groups like "4/7/9" were entirely overlapping, and could only be separated using higher eigenvectors (Figure 1, insets). On the other side of the spectrum, exaggeration values $0 < \rho < 1$ resulted in inflated coalescing clusters.

Interestingly, we observed that FA2 produced an embedding nearly identical to the one at $\rho = 30$, while UMAP produced an embedding nearly identical to the one at $\rho = 4$. Decreasing the $\gamma$ value in UMAP had the same effect as increasing the $\rho$ in t-SNE, and moved the UMAP result towards the LE part of the spectrum (Figure S2).

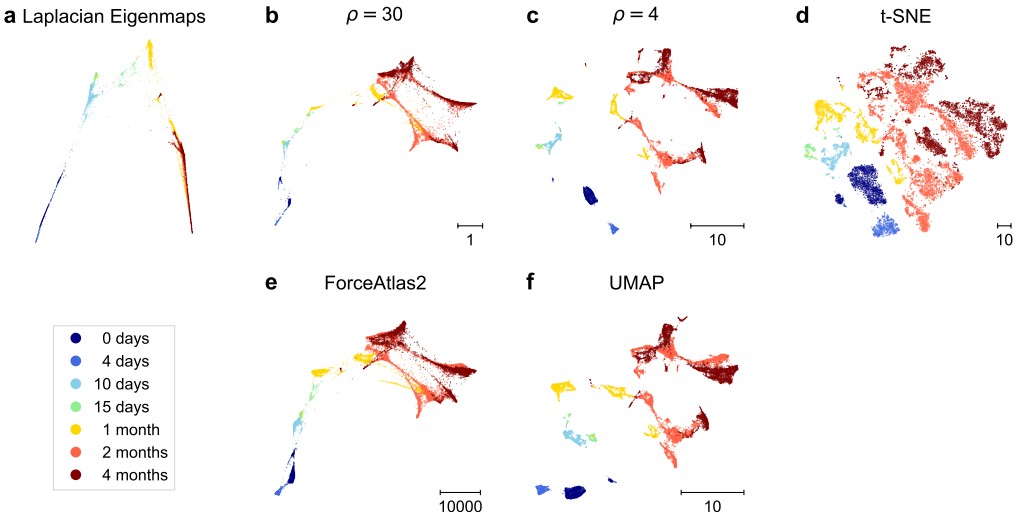

Figure 3: **Neighbor embeddings of the single-cell RNA-seq developmental data.** Cells were sampled from human brain organoids (cell line `409b2`) at seven time points between 0 days and 4 months into the development (Kanton et al., 2019). Sample size $n = 20\,272$. Data were reduced with PCA to 50 dimensions. See Appendix for transcriptomic data preprocessing steps.

The MNIST example suggests that high attraction emphasizes connections between clusters at the cost of within-cluster structure, whereas high repulsion emphasizes the cluster structure at the expense of between-cluster connections. We interpreted this finding as a *continuity-discreteness trade-off*. We developed a simple toy example to illustrate this trade-off in more detail (Figure 2). For this, we generated data as draws from 20 standard isotropic Gaussians in 50 dimensions, each shifted by 6 standard deviation units from the previous one along one axis (1000 points per Gaussian, so $n = 20\,000$ overall). For this analysis we used random initialization and turned the early exaggeration off, to isolate the effect of each loss function on the 'unwrapping' of the random initial configuration.

We found that t-SNE with strong exaggeration ($\rho = 30$) recovered the underlying one-dimensional manifold structure of the data almost as well as LE (Figure 2a,b), and produced an embedding very similar to that of FA2 (Figure 2e). In both cases, the individual clusters were almost invisible. In contrast, embeddings with weaker attraction and stronger repulsion (t-SNE with exaggeration $\rho = 2$ and UMAP) showed individual clusters but were unable to fully recover the 1-dimensional structure and only found some chunks of it (Figure 2c,f). Finally, standard t-SNE clearly showed 20 individual clusters but with the continuous structure entirely lost (Figure 2d).

Further, we analyzed a developmental single-cell transcriptomic dataset, where cells were collected from human brain organoids at seven time points between 0 days and 4 months into the development (Kanton et al., 2019). In this kind of data, one expects to find rich cluster structure as well as a strong time-dependent trajectory. As in the other datasets, we found that stronger attraction (LE, FA2, t-SNE with $\rho = 30$) better represented the developmental trajectory, whereas stronger repulsion (standard t-SNE) better represented the cluster structure (Figure 3). Using much higher $k$ for the kNN graph construction made the developmental trajectory in high-attraction methods even clearer (Figure S3), in agreement with the FA2-based analysis performed in the original publication. We observed the same pattern in a separate dataset obtained from chimpanzee brain organoids (Figures S4, S5).

To quantify our observations, we computed distance correlations (Szekely et al., 2007) between UMAP & FA2 embeddings and t-SNE embeddings with various values of $\rho \in [1, 100]$, using the three datasets analyzed above (Figure 4a). For UMAP, the best matches were at $3 < \rho < 8$; for FA2, the best matches were at $25 < \rho < 65$. The corresponding correlations were all above 0.97, indicating very similar layouts.

We confirmed all these findings using six other datasets (Table 1): Fashion MNIST (Figure S6), Kannada MNIST (Figure S7), Kuzushiji MNIST (Figure S8), single-cell data from a hydra (Figure S9), from a zebrafish embryo (Figure S10), and from a mouse cortex (Figure S11). In most cases, UMAP

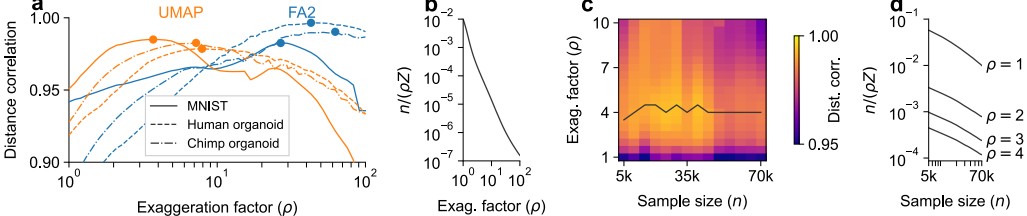

Figure 4: **(a)** Distance correlation (Szekely et al., 2007) between UMAP/FA2 and t-SNE with various exaggeration values $\rho \in [1, 100]$ (50 evenly distributed values on a log scale). Distance correlation was computed using `dcor` package (Carreño, 2017) on a random subset ($n = 5\,000$) of data. Dots mark the maximum of each curve. **(b)** The $n/(\rho Z)$ factor in the end of optimisation when using t-SNE with $\rho \in [1, 100]$ on full MNIST. **(c)** Distance correlations between t-SNE with $\rho \in [1, 10]$ and UMAP depending on the sample size, for MNIST subsets of size $n \in [5\,000, 70\,000]$. Black line indicates best matching $\rho$ values. **(d)** The $n/(\rho Z)$ factor in the end of optimisation when using t-SNE with $\rho \in \{1, 2, 3, 4\}$ on MNIST subsets of size $n \in [5\,000, 70\,000]$.

was similar to t-SNE with exaggeration $\rho \approx 4$ and FA2 was similar to t-SNE with exaggeration $\rho \approx 30$. Increasing attraction strength made the continuous structures increasingly prominent in the embeddings. See Figure S11 for the case when kNN graph had disconnected components.

## 5   INCREASED ATTRACTION IN UMAP DUE TO NEGATIVE SAMPLING

As shown above, the gradient of UMAP (Eq. 9) is very similar to the gradient of t-SNE (Eq. 4) but does not contain the 'normalizing' $n/Z$ term in front of the repulsive forces. What are the typical values of this coefficient? The $Z$ term in t-SNE evolves during optimisation: it starts at $Z \approx n^2$ due to all $d_{ij} \approx 0$ at initialization and decreases towards $n$ as the embedding expands (for a perfect embedding with all $p_{ij} = q_{ij}$ and $v_{ij} = w_{ij}$, the $Z$ would equal $n$; in reality $Z$ usually still exceeds $n$). For MNIST, the final $Z$ value was $\sim 100n$, corresponding to the final $n/Z \approx 0.01$ (Figure 4b). Increasing the exaggeration shrinks the embedding and increases the final $Z$; it also changes the repulsive factor to $n/(\rho Z)$ (Eq. 5). For $\rho = 4$, the final $Z$ was $\sim 2100n$, corresponding to final $n/(\rho Z) \approx 0.0001$ (Figure 4b). This means that UMAP matched t-SNE results with the repulsive factor 0.0001 better than it matched t-SNE results with the repulsive factor 0.01, even though UMAP itself uses repulsive factor $\gamma = 1$ (Eq. 9). How is this possible?

We hypothesized that this mismatch arises because the UMAP implementation is based on negative sampling and does not in fact optimize its stated loss (Eq. 7). Instead, the negative sampling decreases the repulsion strength, creating an effective $\gamma_{\text{eff}}(\nu) \ll 1$. We verified that increasing the value of $\nu$

Table 1: Distance correlation values ($\mathcal{R}$) between UMAP/FA2 and t-SNE.

| Dataset | UMAP | | | FA2 | | |
|---|---|---|---|---|---|---|
| | $\rho = 4$ | $\max \mathcal{R}$ | $\arg\max_\rho \mathcal{R}$ | $\rho = 30$ | $\max \mathcal{R}$ | $\arg\max_\rho \mathcal{R}$ |
| MNIST | 0.985 | 0.985 | 3.7 | 0.982 | 0.983 | 26.8 |
| Fashion MNIST | 0.991 | 0.996 | 7.2 | 0.992 | 0.995 | 75.4 |
| Kuzushiji MNIST | 0.944 | 0.95 | 12.6 | 0.974 | 0.982 | 68.7 |
| Kannada MNIST | 0.936 | 0.952 | 11.5 | 0.976 | 0.978 | 12.6 |
| Chimpanzee organoid | 0.979 | 0.983 | 7.2 | 0.988 | 0.993 | 62.5 |
| Human organoid | 0.972 | 0.979 | 7.9 | 0.995 | 0.997 | 51.8 |
| Hydra | 0.98 | 0.986 | 16.8 | 0.981 | 0.993 | 22.2 |
| Mouse cortex | 0.95 | 0.957 | 12.6 | 0.985 | 0.986 | 29.5 |
| Zebrafish embryo | 0.974 | 0.982 | 8.7 | 0.995 | 0.996 | 42.9 |

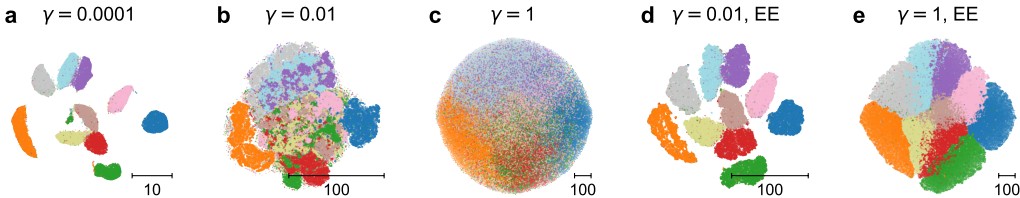

Figure 5: **Barnes-Hut UMAP without negative sampling. (a–c)** Embeddings with gamma values $\gamma \in \{0.0001, 0.01, 1\}$. **(d–e)** Embeddings with gamma values $\gamma \in \{0.01, 1\}$ initialized with the embedding with $\gamma = 0.0001$ [panel (a)], in an analogy to early exaggeration in t-SNE.

increased the repulsion strength in UMAP (Figure S12): embeddings grew in size and the amount of between-cluster white space decreased. We observed the same effect when setting $\epsilon = 1$ (Figure S13), which makes the UMAP gradient (Eq. 9) equivalent to the t-SNE gradient, up to the repulsion factor.

It is difficult to compute $\gamma_{\text{eff}}(\nu)$ analytically, but qualitatively the number of repulsive forces per one attractive force is $\sim n/k$ in the full gradient but $\nu$ with negative sampling. This suggests that the $\gamma_{\text{eff}}$ induced by the negative sampling should decrease with the sample size. To check this prediction, we looked at the behaviour of the $n/(\rho Z)$ term as a function of sample size, for the best matching $\rho$ value. Using MNIST subsets of different sizes $n \in [5\,000, 70\,000]$, we found that $\rho \approx 4$ gave the t-SNE embedding best matching to the UMAP embedding for all considered sample sizes (Figure 4d). Looking now at the final $n/(\rho Z)$ values with $\rho = 4$, we found that they decreased with $n$ approximately as $\sim \mathcal{O}(1/\sqrt{n})$ (Figure 4d), qualitatively confirming our prediction about $\gamma_{\text{eff}}$.

To confirm our interpretation, we developed a Barnes-Hut UMAP implementation (based on `openTSNE`) that optimizes the full UMAP loss without any negative sampling (Figure 5). On full MNIST, $\gamma = 0.0001$ yielded an embedding that resembled the standard (negative-sampling-based) UMAP (Figure 5a), while larger values of $\gamma$ yielded over-repulsed embeddings (Figure 5b–c) and required early exaggeration to produce meaningful results (Figure 5d–e), with $\gamma = 0.01$ resembling t-SNE and $\gamma = 1$ being over-repulsed compared to t-SNE. This suggests that directly optimizing the cross-entropy loss (Eq. 7) is not a viable NE strategy. Therefore, the more condensed clusters typically observed in UMAP compared to t-SNE are an accidental by-product of UMAP's negative sampling strategy, and not a consequence of the cross-entropy loss function itself or the mathematical apparatus of the original paper (McInnes et al., 2018).

## 6    INCREASED ATTRACTION IN FA2 DUE TO NON-DECAYING ATTRACTIVE FORCES

The attractive forces in t-SNE scale as $d_{ij}/(1 + d_{ij}^2)$. When all $d_{ij}$ are small, this becomes an approximately linear dependency on $d_{ij}$, which is the reason why t-SNE with high exaggeration $\rho \gg 1$ replicates Laplacian eigenmaps (see Section 3.2 and Appendix). For large distances $d_{ij}$, attractive forces in t-SNE decay to zero, making default t-SNE very different from LE. In contrast, in FA2, attractive forces always scale as $d_{ij}$. Thus, the larger the embedding distance between points, the stronger the attractive force between them. This strong non-decaying attractive force makes FA2 behave similar to Laplacian eigenmaps on the attraction-repulsion spectrum.

Note that it is not possible to move FA2 embeddings along the attraction-repulsion spectrum by multiplying the attractive or repulsive forces by a constant factor (such as $\gamma$ in UMAP or $\rho$ in t-SNE). Multiplying attractive forces by any factor $a$ or repulsive forces by any factor $1/a$ only leads to rescaling of the embedding by $1/\sqrt{a}$. Indeed, if all forces are in equilibrium before such multiplication and rescaling, they will stay in equilibrium afterwards. This is a general property of force-directed layouts where both attractive and repulsive forces scale as powers of the embedding distance. This argument also implies that removing the $(h_i + 1)(h_j + 1) \approx k^2$ factor from the FA2 gradient (Eq. 11) rescales the entire embedding by $\sim 1/k$, but does not change it otherwise (Figure S1).

# 7 DISCUSSION

We showed that changing the balance between attractive and repulsive forces in t-SNE directly affects the trade-off between preserving continuous or discrete structures. Increasingly strong repulsion 'brings out' information from higher Laplacian eigenvectors into the two embedding dimensions (Figure 1). It is remarkable that the repulsive forces, which are data-agnostic and do not depend on the input data (Carreira-Perpinán, 2010), have so much qualitative influence.

Our results suggest that it should be beneficial for high-repulsion embeddings to begin optimization with lower repulsion strength, in order to better preserve global structure. This explains how UMAP benefits from its default initialization with Laplacian eigenmaps (Kobak & Linderman, 2019) and how t-SNE benefits from early exaggeration (Linderman & Steinerberger, 2019) (see Figure S14 for a demonstration of the importance of early exaggeration). Similarly, elastic embedding gradually increases repulsion strength during optimisation (Carreira-Perpinán, 2010).

Our treatment provided a unified perspective on several well-known NE algorithms that have scalable implementations and that have been shown to successfully embed datasets such as MNIST without coarse-graining the kNN graph. Methods based on coarse-graining, such as e.g. PHATE (Moon et al., 2019) or latent variable NE method in Saul (2020) may behave differently. We believe that our treatment may allow to position other NE algorithms on the same spectrum. For example, a recently suggested TriMap algorithm (Amid & Warmuth, 2019), which uses negative sampling similar to UMAP, appears to have stronger attractive forces than UMAP (cf. Figure 5 in the original paper), with some TriMap embeddings, e.g. of the Fashion MNIST dataset, looking similar to the ForceAtlas2 embeddings shown in our work.

We argued that negative sampling (Mikolov et al., 2013) used by LargeVis/UMAP strongly lowers the effective repulsion. Negative sampling is closely related to the *noise-contrastive estimation* (NCE) framework (Gutmann & Hyvärinen, 2012). NCE was recently applied to t-SNE under the name of NCVis (Artemenkov & Panov, 2020), and the general NCE theory asserts that it should be asymptotically equivalent to optimizing the full gradient (Gutmann & Hyvärinen, 2012). We consider it a very interesting research direction to study the relationship between negative sampling and NCE and their effect on 2D embeddings as well as on higher-dimensional embeddings used in methods like `word2vec` (Mikolov et al., 2013).

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

## APPENDIX

### APPENDIX A. RELATIONSHIP TO LAPLACIAN EIGENMAPS

**Laplacian eigenmaps**    Let a $n \times n$ symmetric matrix $\mathbf{V}$ contain pairwise affinities between $n$ points (or edge weights between nodes in an undirected graph). Let diagonal matrix $\mathbf{D}$ contain row (or, equivalently, column) sums of $\mathbf{V}$, i.e. $D_{ii} = \sum_j V_{ij}$. Then $\mathbf{L} = \mathbf{D} - \mathbf{V}$ is known as (unnormalized) graph Laplacian, and Laplacian eigenmaps (Belkin & Niyogi, 2002) can be formulated as solving the generalized eigenvector problem

$$\mathbf{La} = \lambda \mathbf{Da} \tag{12}$$

and taking the eigenvectors corresponding to the *smallest* eigenvalues (after discarding the trivial eigenvector $[1, 1, \ldots, 1]^\top$ with eigenvalue zero). By multiplying both sides of this equation by $\mathbf{D}^{-1}$, the problem can be reformulated as finding the eigenvectors of $\mathbf{D}^{-1}\mathbf{V}$ corresponding to the *largest* eigenvectors:

$$\mathbf{D}^{-1}\mathbf{Va} = (1 - \lambda)\mathbf{a}. \tag{13}$$

The matrix $\mathbf{D}^{-1}\mathbf{V}$ is not symmetric and has rows normalized to 1. It can be interpreted as the diffusion operator on the graph, making Laplacian eigenmaps equivalent to *Diffusion maps* (Coifman & Lafon, 2006). Another equivalent way to rewrite it, is to define normalized Laplacian $\mathbf{L}_{\mathrm{norm}} = \mathbf{D}^{-1/2}\mathbf{L}\mathbf{D}^{-1/2}$ and solve an eigenvector problem $\mathbf{L}_{\mathrm{norm}}\mathbf{b} = \lambda\mathbf{b}$, where $\mathbf{b} = \mathbf{D}^{1/2}\mathbf{a}$.

**t-SNE without repulsion**    In the limit of $\rho \to \infty$, the repulsive term in the t-SNE gradient can be dropped, all $w_{ij} \to 1$, and hence the gradient descent update rule becomes (Linderman & Steinerberger, 2019)

$$\mathbf{y}_i^{t+1} = \mathbf{y}_i^t - \eta \sum_j v_{ij}(\mathbf{y}_i^t - \mathbf{y}_j^t), \tag{14}$$

where $t$ indexes the iteration number and $\eta$ is the learning rate (including all constant factors in the gradient). Denoting by $\mathbf{Y}$ the $n \times 2$ matrix of the embedding coordinates, this can be rewritten as

$$\mathbf{Y}^{t+1} = (\mathbf{I} - \eta\mathbf{D} + \eta\mathbf{V})\mathbf{Y}^t \tag{15}$$
$$= \mathbf{MY}^t. \tag{16}$$

$\mathbf{M}$ is the transition matrix of this Markov chain (note that it is symmetric and its rows and columns sum to 1; its values are all non-negative for small enough $\eta$). According to the general theory of Markov chains, the largest eigenvalue of $\mathbf{M}$ is 1, and the corresponding eigenvector is $[1, 1, \ldots, 1]^\top$, meaning that the embedding shrinks to a single point (as expected without repulsion). The slowest shrinking eigenvectors correspond to the next eigenvalues. This means that when $\rho \to \infty$, the embedding (if rescaled, to avoid shrinking to zero) will converge to the leading nontrivial eigenvectors of $\mathbf{M}$. This becomes equivalent to a power iteration algorithm. The eigenvectors of $\mathbf{M}$ are the same as of $\mathbf{L} = \mathbf{D} - \mathbf{V}$, which is the unnormalized graph Laplacian of the symmetric affinity matrix.

Note that this is not precisely what LE computes: as explained above, it finds eigenvectors of the *normalized* graph Laplacian (c.f. Von Luxburg et al., 2008). However, in practice $\mathbf{D}$ is often approximately proportional to the identity matrix, because $\mathbf{V}$ is obtained via symmetrization of directed affinities, and those have rows summing to 1 by construction. We can therefore expect that the leading eigenvectors of $\mathbf{L}$ and of $\mathbf{L}_{\mathrm{sym}}$ are not too different. We verified that for MNIST data they were almost exactly the same.

Note also that nothing prevents different columns of $\mathbf{Y}$ to converge to the same leading eigenvector: each column independently follows its Markov chain. Indeed, we observed that for large enough values of $\rho$ and large enough number of gradient descent iterations, the rescaled two-dimensional embedding collapsed to one dimension. This is the expected limiting behaviour when $\rho \to \infty$. However, for moderate values of $\rho$ (as shown in this manuscript), this typically does not happen, and columns of $\mathbf{Y}$ resemble the two leading non-trivial eigenvectors of the Laplacian. The repulsive force prevents the embedding from collapsing to the leading Laplacian eigenvector. At the same time, a weak repulsive force will only be able to 'bring out' the second LE eigenvector. The stronger the contribution of repulsive forces, the more LE eigenvectors it would be able to 'bring out' (remember that the attractive force acts stronger on the higher eigenvectors).

**Loss function of LE** The original Laplacian eigenmaps paper (Belkin & Niyogi, 2002) motivated the eigenvector problem by considering

$$\mathcal{L}_{\text{LE}} = \sum_{i,j} v_{ij} \|\mathbf{y}_i - \mathbf{y}_j\|^2 = 2 \operatorname{Tr}(\mathbf{Y}^\top \mathbf{L} \mathbf{Y}). \tag{17}$$

This expression can be trivially minimized by setting all $\mathbf{y}_i = \mathbf{0}$, so the authors introduced a quadratic constraint $\mathbf{Y}^\top \mathbf{D} \mathbf{Y} = \mathbf{I}$, yielding the generalized eigenvector problem. We note that a different quadratic constraint $\mathbf{Y}^\top \mathbf{Y} = \mathbf{I}$ would yield a simple eigenvector problem for $\mathbf{L}$. In any case, the constraint plays the role of the repulsion in t-SNE framework.

APPENDIX B. DATA SOURCES AND TRANSCRIPTOMIC DATA PREPROCESSING

**Transcriptomic datasets** The brain organoid datasets (Kanton et al., 2019) were downloaded from `https://www.ebi.ac.uk/arrayexpress/experiments/E-MTAB-7552/` in form of UMI counts and metadata tables. The metadata table for the chimpanzee dataset was taken from the supplementary materials of the original publication. We used gene counts mapped to the consensus genome, and selected all cells that passed quality control by the original authors (`in_FullLineage=TRUE` in metadata tables). For human organoid data, we only used cells from the 409b2 cell line, to simplify the analysis (the original publication combined cells from two cell lines and needed to perform batch correction).

The hydra dataset (Siebert et al., 2019) (Figure S9) was downloaded in form of UMI counts from `https://www.ncbi.nlm.nih.gov/geo/query/acc.cgi?acc=GSE121617`.

The zebrafish dataset (Wagner et al., 2018b) (Figure S10) was downloaded in form of UMI counts from `https://kleintools.hms.harvard.edu/paper_websites/wagner_zebrafish_timecourse2018/WagnerScience2018.h5ad`.

The adult mouse cortex dataset (Tasic et al., 2018) (Figure S11) was downloaded in form of read counts from `http://celltypes.brain-map.org/api/v2/well_known_file_download/694413985` and `http://celltypes.brain-map.org/api/v2/well_known_file_download/694413179` for the VISp and ALM cortical areas, respectively. Only exon counts were used here. The cluster labels and cluster colors were retrieved from `http://celltypes.brain-map.org/rnaseq/mouse/v1-alm`.

To preprocess each dataset, we selected 1000 most variable genes using procedure from Kobak & Berens (2019) with default parameters (for the mouse cortex dataset we used 3000 genes and `threshold=32` (Kobak & Berens, 2019)) and followed the preprocessing pipeline from the same paper: normalized all counts by cell sequencing depth (sum of gene counts in each cell), multiplied by the median cell depth (or 1 million in case of mouse cortex data), applied $\log_2(x+1)$ transformation, did PCA, and retained 50 leading PCs.

**MNIST-like datasets** The datasets used in Figures S6, S7, and S8 have been published explicitly to function as drop in replacements for the handwritten MNIST dataset. The dataset variants that we used here all consist of a total $n = 70\,000$ images of $28 \times 28$ pixels, in 10 balanced classes. The input has been preprocessed like the original MNIST dataset, i.e. reduced to 50 dimensions via PCA. Fashion and Kuzushiji MNIST were downloaded via OpenML with the keys `Fashion-MNIST` (`https://www.openml.org/d/40996`) and `Kuzushiji-MNIST` (`https://www.openml.org/d/41982`), respectively. Kannada MNIST was downloaded from `https://github.com/vinayprabhu/Kannada_MNIST`.

## SUPPLEMENTARY FIGURES

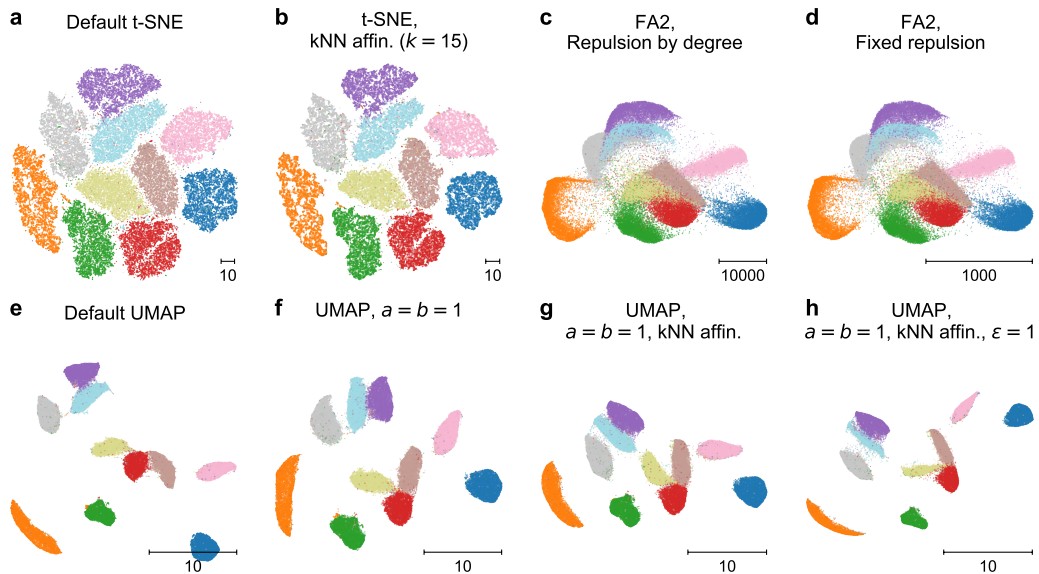

Figure S1: **Different algorithmic choices, demonstrated using MNIST. (a)** Default t-SNE embedding, perplexity 30. **(b)** T-SNE embedding with binary kNN affinities, $k = 15$. All non-zero affinities have equal size. The $p_{ij}$ values were normalized as always in t-SNE, and sum to one. **(c)** Default ForceAtlas2, using binary symmetrized kNN adjacency matrix ($k = 15$) as input. ForceAtlas2 uses so-called repulsion by degree by default. **(d)** ForceAtlas2 without repulsion by degree. **(e)** Defaut UMAP embedding. This uses default values for $a$ and $b$ parameters, and LE initialization. **(f)** UMAP embedding with PCA initialization and Cauchy kernel ($a = b = 1$). **(g)** UMAP embedding with PCA initialization, Cauchy kernel, and binary kNN affinities ($k = 15$). **(h)** UMAP embedding with PCA initialization, Cauchy kernel, binary kNN affinities, and $\epsilon = 1$.

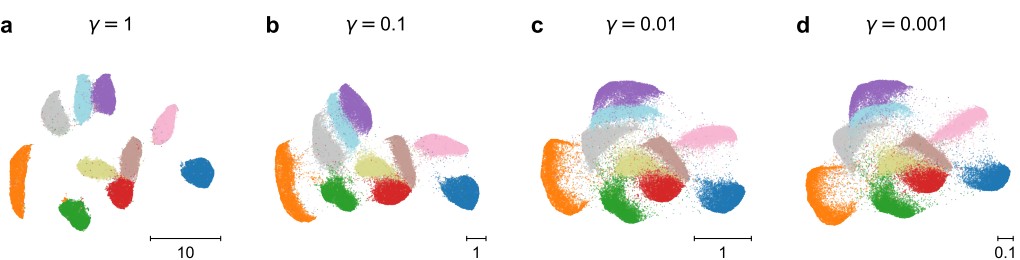

Figure S2: **Decreasing the repulsion in UMAP. (a)** UMAP embedding of MNIST with $\gamma = 1$ (default). **(b–d)** Decreasing $\gamma$ produces the same effect as increasing the exaggeration $\rho$ in t-SNE. Values $\gamma > 1$ are not shown because we found that it was difficult to achieve a well-converged embedding for $\gamma \gg 1$.

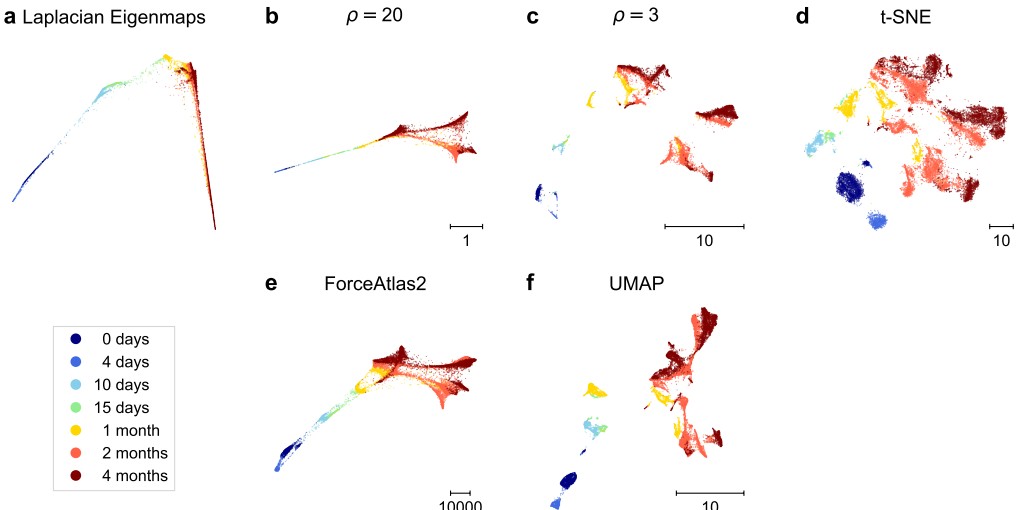

Figure S3: **Neighbor embeddings of the single-cell RNA-seq developmental data (human, high** $k$**).** Cells were sampled from human brain organoids (cell line `409b2`) at seven time points between 0 days and 4 months into the development (Kanton et al., 2019). Sample size $n = 20\,272$. Data were reduced with PCA to 50 dimensions. See Appendix for transcriptomic data preprocessing steps. LE, FA2, and UMAP used $k = 150$ (instead of our default $k = 15$), while t-SNE used perplexity 300 (instead of our default 30).

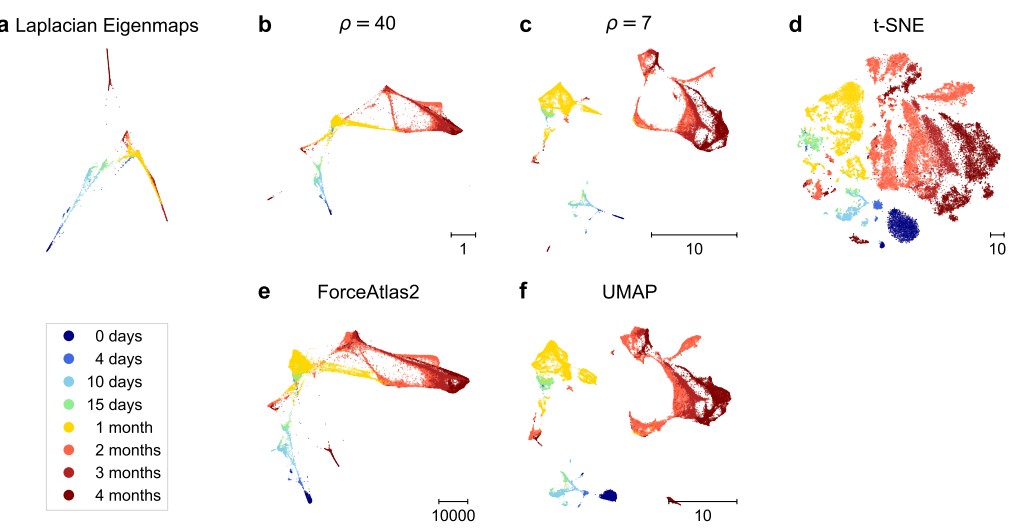

Figure S4: **Neighbor embeddings of the single-cell RNA-seq developmental data (chimpanzee).** Cells were sampled from chimpanzee brain organoids at eight time points between 0 days and 4 months into the development (Kanton et al., 2019). Sample size $n = 36\,884$. Data were reduced with PCA to 50 dimensions. See Appendix for transcriptomic data preprocessing steps.

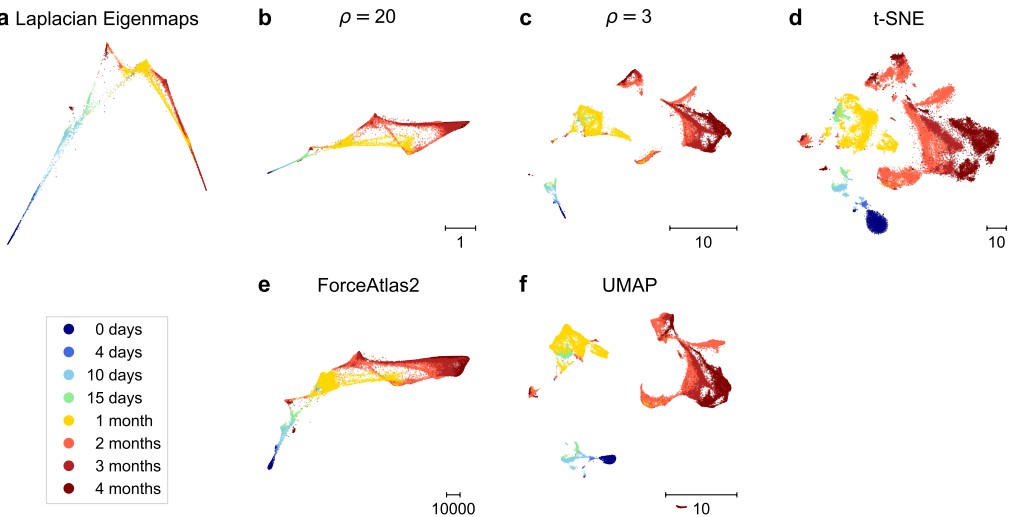

Figure S5: **Neighbor embeddings of the single-cell RNA-seq developmental data (chimpanzee, high $k$).** Cells were sampled from chimpanzee brain organoids at eight time points between 0 days and 4 months into the development (Kanton et al., 2019). Sample size $n = 36\,884$. Data were reduced with PCA to 50 dimensions. See Appendix for transcriptomic data preprocessing steps. LE, FA2, and UMAP used $k = 150$ (instead of our default $k = 15$), while t-SNE used perplexity 300 (instead of our default 30).

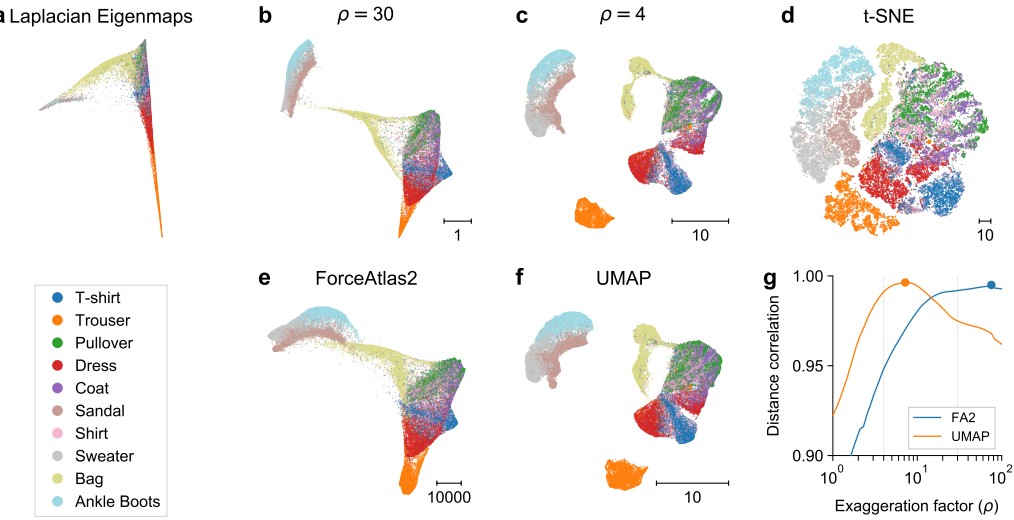

Figure S6: **Fashion MNIST dataset (Xiao et al., 2017).** Sample size $n = 70\,000$. Dimensionality was reduced to 50 with PCA. Colors correspond to 10 classes, see legend.

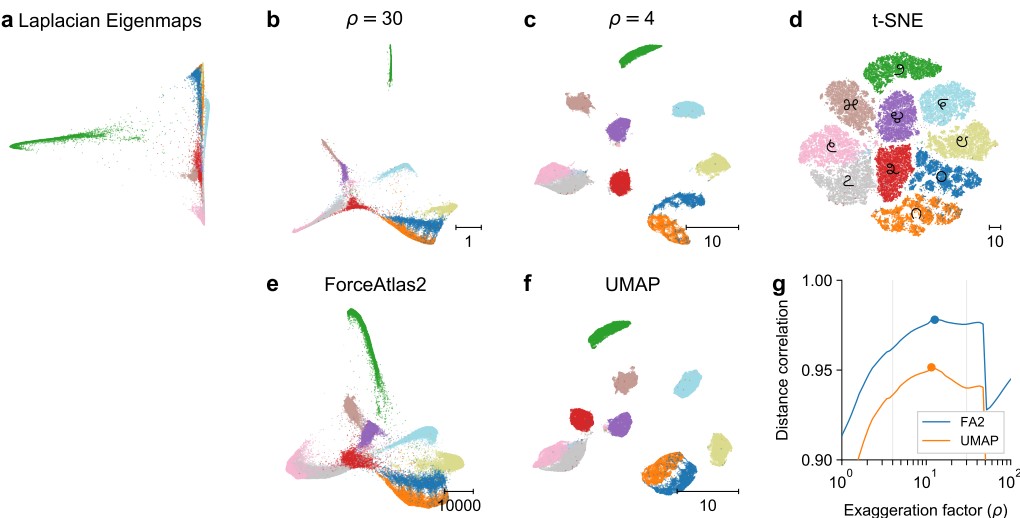

Figure S7: **Kannada MNIST dataset Prabhu (2019).** Sample size $n = 70\,000$. Dimensionality was reduced to 50 with PCA. Colors correspond to 10 Kannada digits shown in panel (d).

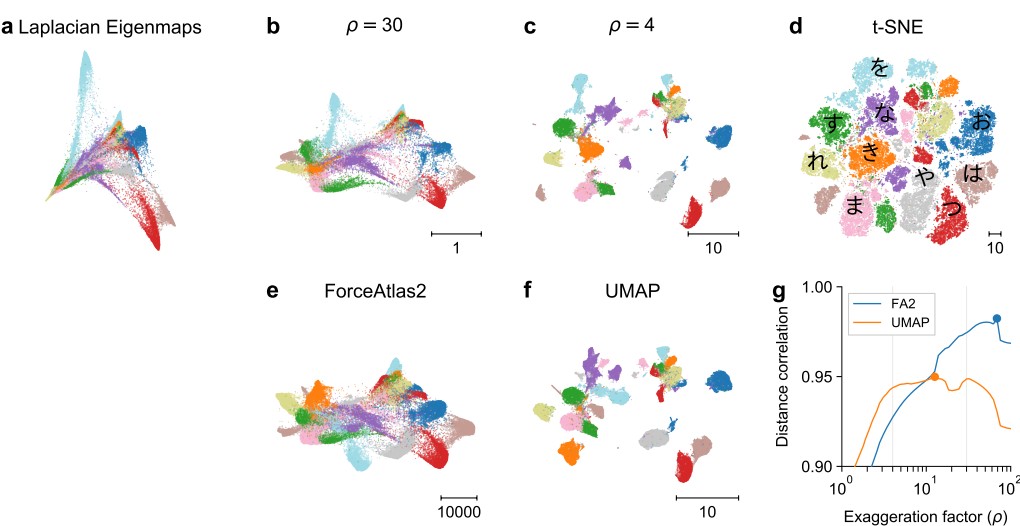

Figure S8: **Kuzushiji MNIST dataset (Clanuwat et al., 2018).** Sample size $n = 70\,000$. Dimensionality was reduced to 50 with PCA. Colors correspond to 10 Kanji characters shown in panel (d).

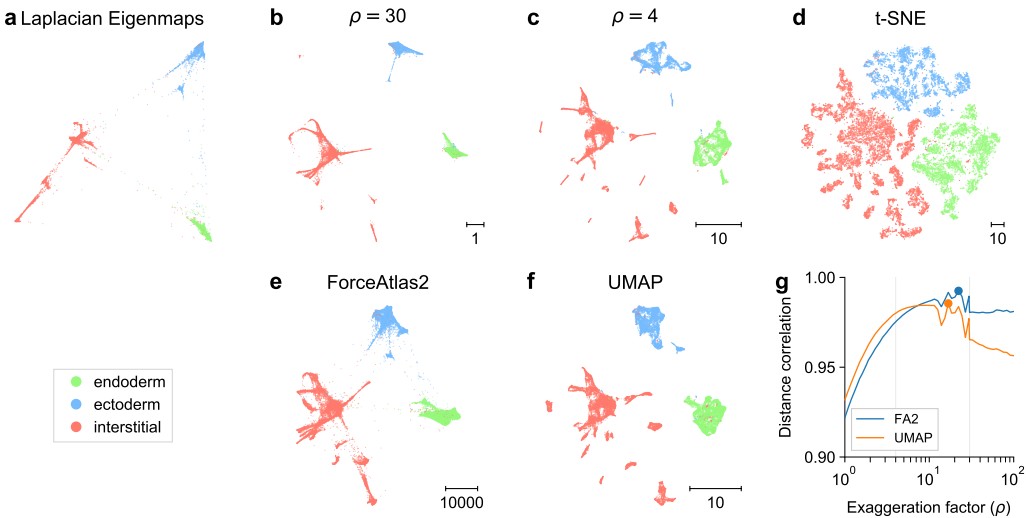

Figure S9: **Single-cell RNA-seq data of a hydra (Siebert et al., 2019).** Sample size $n = 24\,985$. Dimensionality was reduced to 50 with PCA. See Appendix for transcriptomic data preprocessing steps. Color corresponds to cell classes.

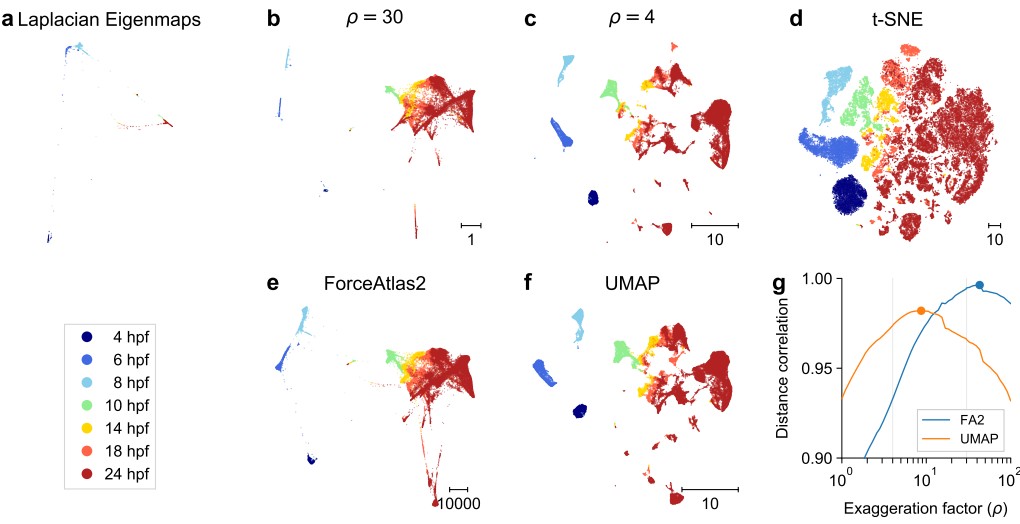

Figure S10: **Single-cell RNA-seq data of a zebrafish embryo (Wagner et al., 2018b).** Sample size $n = 63\,530$. Dimensionality was reduced to 50 with PCA. See Appendix for transcriptomic data preprocessing steps. Color corresponds to the developmental stage, indicating the hours post fertilization (hpf).

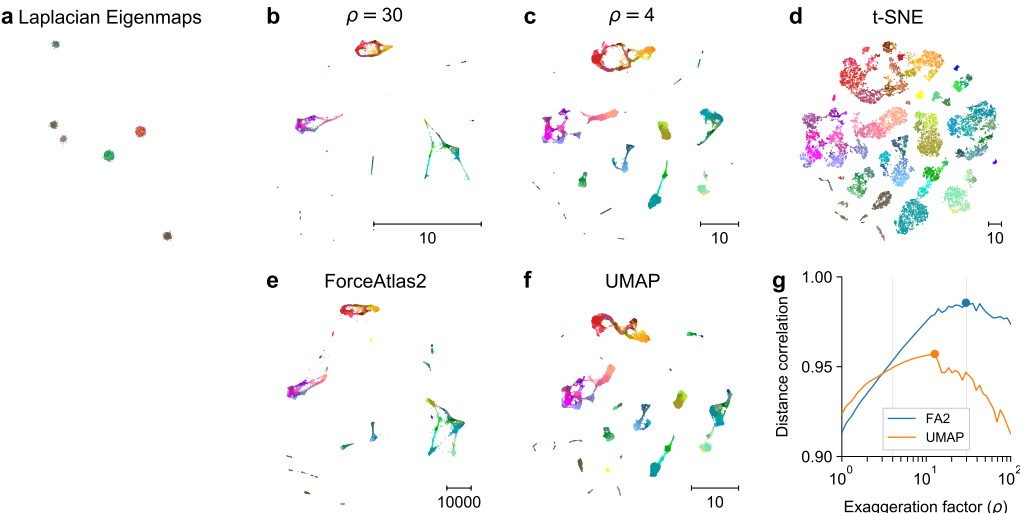

Figure S11: **Single-cell RNA-seq data of adult mouse cortex (**Tasic et al., 2018**).** Sample size $n = 23\,822$. Dimensionality was reduced to 50 with PCA. See Appendix for transcriptomic data preprocessing steps. Colors are taken from the original publication (warm colors: inhibitory neurons; cold colors: excitatory neurons; grey/brown: non-neural cells). We added Gaussian noise to the LE embedding in panel (a) to make the clusters more visible. In this dataset, the kNN graph is disconnected and has 6 components, resulting in 6 distinct points in the LE embedding.

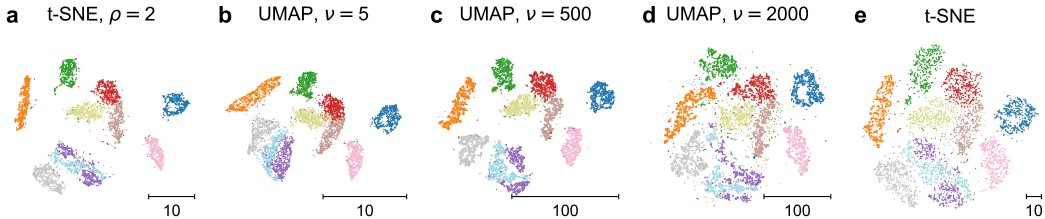

Figure S12: **The effect of negative sampling rate on UMAP embeddings. (a)** T-SNE embedding with $\rho = 2$ of a MNIST subsample with $n = 6000$. **(b–d)** UMAP embeddings with $\nu \in \{5, 500, 2000\}$. We used a $n = 6000$ subset of MNIST as it would take prohibitively long to run optimisation with high $\nu$ on the full MNIST dataset. We increased the number of iterations to ensure convergence (`n_epochs=3000`) and initialized UMAP runs with the default UMAP embedding (in an analogy to early exaggeration in t-SNE). **(e)** Standard t-SNE of the same data.

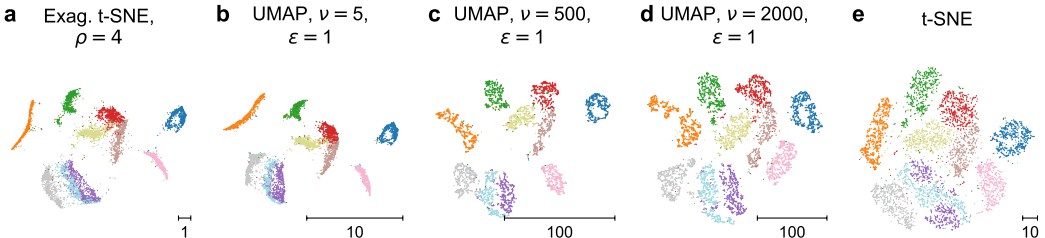

Figure S13: **The effect of negative sampling rate on UMAP embeddings with $\epsilon = 1$. (a)** t-SNE embedding with $\rho = 4$ of a MNIST subsample with $n = 6000$. **(b–d)** UMAP embeddings with $\nu \in \{5, 500, 2000\}$ and $\epsilon = 1$. Here UMAP was run for 3000 epochs to ensure convergence, and was initialized with the default UMAP embedding ($\nu = 5$ with 500 epochs). **(e)** Standard t-SNE of the same data.

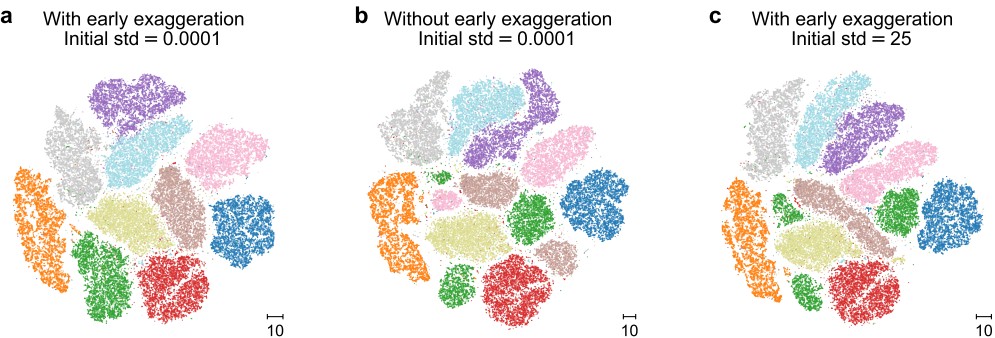

Figure S14: **The effect of early exaggeration on t-SNE**. **(a)** Default t-SNE embedding of MNIST. This uses early exaggeration and sets the standard deviation of PCA initialization to 0.0001. **(b)** T-SNE embedding without early exaggeration. This embedding is stuck in a suboptimal local minimum with some clusters split into multiple parts. **(c)** T-SNE embedding with early exaggeration, but with initial standard deviation set to 25. The attractive forces are too weak to pull the clusters together during the early exaggeration phase.

