# OpenReview forum: "A Unifying Perspective on Neighbor Embeddings along the Attraction-Repulsion Spectrum"
_ICLR.cc/2021/Conference — Reject_

### Official Review · AnonReviewer1 · 2020-10-21
**A unified perspective based on the attraction-replusion spectrum is proposed to empirically study various neighbor embedding methods.**

**Rating:** 5
**Confidence:** 3

**Review:**

Authors of this paper conducted study on existing neighbor embedding methods from the perspective of the attraction-replusion spectrum.

Pros:
1)	Extensive experiments demonstrate the phenomena that neighbor embedding methods can achieve the continuous manifold structures or discrete clustering structures by varying the balance parameter.
2)	Detailed implementations of existing methods are taken into account for the analysis, so it is a good guideline for users of these methods on real applications.

Cons:
1)	The novelty of this paper seems limited. The attraction-replusion property is not new to the existing neighbor embedding methods.
2)	Authors conducted various experiments to show the property, but the conclusion is only empirical.
3)	Authors may need to demonstrate how these observations can make improvement over the existing methods or motivate new models.
4)	To better demonstrate the continuous manifold or discrete clustering structures, authors can take various real data with clear underlying manifold structures that have been studied in the literature. Simulation data is good, but it is less real since the noise may significantly affect the experimental results.

---

> ### Author Response · Authors · 2020-11-13
> **Author response**
>
> We thank the reviewer for their review, and for highlighting the Pros of our work, specifically noting the value of our extensive experiments. However, we disagree with most of the listed Cons, and will respond to each of them below. In light of our replies, we would be glad if the reviewer would reevaluate their assessment.
>
> 1. The novelty of our paper is in two parts. First, we show that varying the exaggeration in t-SNE leads to the continuity-discreteness tradeoff. Whereas “early exaggeration” has been discussed and analyzed before in the exaggeration->∞ limit, our results about spectrum and tradeoff are entirely novel. But second, even more importantly, we explain how UMAP and ForceAtlas2 fall on this spectrum, and this is 100% new results with no prior research whatsoever.
>
>   Perhaps it will be helpful to give some context here. t-SNE and UMAP (and to a smaller extent ForceAtlas2) are *immensely* popular methods in the single-cell transcriptomic community. Hundreds and thousands of biology papers use these methods to visualize single cell data. Recently, especially after Becht et al 2018 (Nat Biotechnology), the “common knowledge” in transcriptomic and also in manifold learning community is that UMAP is better suited for single cell analysis because its cross-entropy loss function is more adequate than the Kullback-Leibler loss function of t-SNE. There are numerous articles and popular blog posts that make this claim. Our work shows that this is simply not the case; in fact, UMAP works *despite* the cross-entropy loss (because negative sampling effectively lowers the repulsion by several orders of magnitude!). This is an entirely novel statement that we are certain must be surprising to many in the field.
>
>   Similarly, ForceAtlas2 is widely used in the transcriptomic community (see references in our paper) to highlight continuous developmental trajectories, and there has been no theoretical explanation for why it would work like this. Our paper resolves this puzzle.
>
> 2. Several reviewers raised the concern that our conclusions are “only empirical” but we e.g. present a mathematical analysis of UMAP optimization, prove that it effectively decreases the repulsion, and then re-implement UMAP using Barnes-Hut to demonstrate this experimentally. We do not understand how this can be described as “only empirical”.
>
> 3. The focus of this work was not in developing new methods, so the reviewer is right that this paper does not suggest any. Our goal was to provide a unifying framework to think about neighbor embeddings. We certainly hope that this can motivate further method development, e.g. by highlighting the continuity-discreteness tradeoffs between existing methods and opening the questions of how to preserve both types of structure in a single embedding, or how to choose which of the embeddings on this spectrum is more adequate to the underlying data.
>
> 4. The reviewer suggests to analyze “various real data [...] that have been studied in the literature” but we feel there must be some confusion here because we used nine (!) different real datasets, five of which are published transcriptomic datasets extensively analyzed in the respective original publications. The main text shows 2 real datasets (Figures 1 and 3) and the Appendix shows 7 more (Figures S3--S11). All the results are summarized in the last paragraph on page 6 in the main text and in the Table S1, and are strongly consistent across datasets.

---

> ### Author Response · Authors · 2020-11-24
> **Revision uploaded**
>
> We have uploaded a revision, where we addressed some of the issues raised by the Reviewers and moved Table S1 into the main text to make our set of experiments on "various real data" appear more prominently. We thank the Reviewer again for their comments.

---

### Official Review · AnonReviewer2 · 2020-10-28
**Interpretation is nice, but what kind of practical benefits can we get?**

**Rating:** 5
**Confidence:** 2

**Review:**

Summary:
In this paper, a unified view of embedding methods for visualization is presented. The main message is that, Laplacian eigenmaps and t-SNE are governed by a single formula, and the difference of them can be seen as a difference of a hyperparameter value. We can also approximately recover two different embedding methods --- UMAP and ForceAtlas2. Using a few benchmark data sets, the relationships of these methods are visualized.


Detailed comments:
First of all, I have almost no experience to study visualization methods, and my assessments, especially towards non-technical parts, could be biased.

The main result --- a general equation that covers several embedding methods --- is technically interesting. However, it would be not clarified what kind of benefits we can get from the finding. A possible direction is to tune $\rho$ so that we can get the most effective visualization. Is it possible to do something like this?

Another concern is that it is not easy to judge the visualization results (e.g. Fig 2) because they are qualitative and subjective. To prove the significance of this study, more quantitative results that everyone can understand would be necessary.


Minor comments:
- $\sim$ in Eq. (3) seems to mean the equality up to constant but such usage would be not common. It would be better to add an explanation.

---

> ### Author Response · Authors · 2020-11-13
> **Author response**
>
> We thank the reviewer for their review. As the reviewer said that they have “almost no experience” in manifold learning and low-dimensional visualization, we would like to clarify the context of our work.
>
> t-SNE and UMAP (and to a smaller extent ForceAtlas2) are *immensely* popular methods in the single-cell transcriptomic community. Hundreds and thousands of biology papers use these methods to visualize single cell data. Recently, especially after Becht et al 2018 (Nat Biotechnology), the “common knowledge” in transcriptomic and also in manifold learning community is that UMAP is better suited for single cell analysis because its cross-entropy loss function is more adequate than the Kullback-Leibler loss function of t-SNE. There are numerous articles and popular blog posts that make this claim. Our work shows that this is simply not the case; in fact, UMAP works *despite* the cross-entropy loss (because negative sampling effectively lowers the repulsion by several orders of magnitude!). This is an entirely novel statement that we are certain must be surprising to many in the field.
>
> Similarly, ForceAtlas2 is widely used in the transcriptomic community (see references in our paper) to highlight continuous developmental trajectories, and there has been no theoretical explanation for why it would work like this. Our paper resolves this puzzle.
>
> The reviewer is right that our paper does not develop any new “practical” method. This was not the goal of this paper. Instead, its goal is to place all these embedding methods into a common framework and explain the tradeoffs between them. The value of our paper is in the insight that it provides into how these methods are related and how methods like UMAP *really* work.
>
> Finally, regarding qualitative and quantitative comparisons: the goal of our paper was emphatically *not* to say which embedding is better and which one is worse (e.g. which of the embeddings shown in Figure 1 is better). The goal of our experiments was to compare embeddings done with different algorithms between each other. And we provide quantitative evidence by using the distance correlation metric (Table S1) supporting our claims that UMAP behaves like t-SNE with exaggeration ~4 while ForceAtlas2 behaves like t-SNE with exaggeration ~30.

---

> ### Author Response · Authors · 2020-11-24
> **Revision uploaded**
>
> We have uploaded a revision, where we fixed the minor issue that the Reviewer raised and, more importantly, moved Table S1 into the main text as Table 1 to make our "quantitative results" appear more prominently. We thank the Reviewer again for their comments.

---

### Official Review · AnonReviewer4 · 2020-10-28
**Study of role of repulsion/attraction trade-off in manifold leaning.**

**Rating:** 4
**Confidence:** 5

**Review:**

The paper studies an early-exagerration constant \rho that trades-off the attraction vs repulsion parts of the manifold learning algorithms at the early steps of optimization.

The paper provides a comparative analysis of objective functions for several manifold learning algorithm through the lights of effect of \rho. It shows that the embedding of UMAP and ForceAtlas2 could be roughly recovered using t-SNE with a special choice of \rho. The paper also proposes an explanation why UMAP works better when initialized from LE and why t-SNE gets better results from early exaggeration. All the analysis is done on a single MNIST dataset .

The paper seem very incremental in nature, since Elastic Embedding (Carreira-Perpiñan, 2010) already studied the effect of \rho in a very similar manner to the findings proposed in the paper. The paper, however, only briefly covers that paper, mentioning the algorithm only in he Discussion section.

The authors added the analysis between t-SNE and most recent UMAP and ForceAtlas2 algorithms, but most of the novelty about the effect of \rho and the connections to LE and early exaggeration are quite apparent. Most importantly, the paper doesn't propose any new algorithm or any improvements beyond insights. Even those are defined very vaguely. For example, it is not clear to me, ho the authors define "high-repulsion embeddings" and how much lower the repulsion strength should be.

Finally, the result on a single MNIST dataset are not very general nor conclusive.

---

> ### Author Response · Authors · 2020-11-13
> **Author response**
>
> We thank the reviewer for their review. Unfortunately, there must have been some misunderstanding because the reviewer says that “All the analysis is done on a single MNIST dataset” whereas in reality we used nine (!) different datasets, plus a simulated dataset. The main text shows 2 real life datasets (Figures 1 and 3) and the Appendix shows 7 more (Figures S3--S11). All the results are summarized in the last paragraph on page 6 in the main text and in the Table S1, and are strongly consistent across datasets. We believe this analysis to be quite exhaustive, and would like to ask the reviewer to reconsider their score in light of this.
>
> Regarding the novelty of our work, we would like to point out three things:
>
> 1) The reviewer may think that the connection between UMAP and t-SNE is “quite apparent”, but in fact the entire manifold learning community + the entire transcriptomic community is convinced that UMAP separates clusters and highlights continuous structures better than t-SNE due to mathematical properties of the cross-entropy loss. There are numerous articles and popular blog posts that make this claim. Our work shows that this is simply not the case; in fact, UMAP works *despite* the cross-entropy loss (because negative sampling effectively lowers the repulsion by several orders of magnitude!). This is an entirely novel statement that we are certain must be surprising to many in the field.
>
> 2) The relationship between t-SNE and ForceAtlas2 has not, to the best of our knowledge, analyzed anywhere at all. At the same time, FA2 is widely used in the transcriptomic community (see references in our paper) to highlight continuous developmental trajectories, and there has been no theoretical explanation whatsoever for why it would work like this. Our paper resolves this puzzle.
>
> 3) We are big fans of the Carreira-Perpiñan (2010) paper, and would be happy to mention it more prominently in the Related Work section. That paper was the first to point out the relationship between SNE (not t-SNE though) with ρ->∞ and LE, and we do cite it in this context in Section 3.2 (page 3) and *not* only in the Discussion, as the reviewer claimed. More importantly though, CP2010 suggested to slowly relax attraction forces, which is an important optimization trick, as we say in the Discussion. However, CP2010 did *not* suggest that different levels of attraction strength can yield separately interesting embeddings, and did *not* show any examples of real datasets where different attraction levels would show different aspects of the data. Moreover, CP2010 says nothing about UMAP or ForceAltas2, which is the entire focus of our paper. In light of that, we were surprised that the reviewer would characterize our manuscript as “incremental”. Note that we are not using exaggeration as “early exaggeration” and keep exaggeration turned on throughout the optimization.

---

> > ### Comment · AnonReviewer4 · 2020-11-20
> > **Reviewer response**
> >
> > Thanks to the authors for the detailed answer.
> >
> > * Figure 3 indeed is done on a e single-cell RNA-seq developmental data, however, it is mentioned only in the caption. I would state this more prominently in the main text somewhere.
> > * UMAP, t-SNE and a general class of force-directed methods are indeed very similar algorithms that work by minimizing the trade-off between attraction and repulsion. Whether it is cross-entropy or KL-divergence, it is quite apparent that all of all them try to balance repulsion/attraction trade off.
> > *  Based on my understanding Carreira-Perpiñan (2010) actually does suggest that different \lambda results in different embedding and Figure 3 from their paper looks remarkably close to the figures presented by the authors. More importantly, Carreira-Perpiñan leave \lambda as a hyperparameter for the user, suggesting that it would them who would need to determine desirable trade-off between attraction and repulsion.

---

> > > ### Author Response · Authors · 2020-11-20
> > > **Regarding Figure 3 and elastic embedding**
> > >
> > > With regard to our Figure 3, there is an entire paragraph on page 6 describing the brain organoid development dataset used in that figure. However, to clear up the misunderstanding, we are going to improve the wording to make it clearer that the dataset is indeed a real world dataset. In addition, please note that two paragraphs below we mention the six (!) other datasets that are presented in the Appendix for space reasons, which are also all real data sets.
> > >
> > > Regarding the Elastic Embedding paper, our reading of Figure 3 and the surrounding text is different. Carreira-Perpinan writes: "This small-λ solution globally unfolds the Swiss roll but shows defects similar to those of spectral methods [...]. But these disappear as λ increases [...]". So we think the paper clearly argues that large lambdas work the best and does not discuss the tradeoff that we focus on. Also, please note that the data in that Figure 3 is a small two-dimensional toy dataset with no cluster structure, whereas we analyze multiple high-dimensional real datasets with lots of structure. We are happy to extend our discussion of the EE paper also in the "related work" section, but we continue to maintain that our findings are novel and in particular go beyond the EE paper.
> > >
> > > We will upload a revision on Monday. Thank you.

---

> ### Author Response · Authors · 2020-11-24
> **Revision uploaded**
>
> We have uploaded a revision, mentioning the Elastic Embedding paper in the related work section and moving Table S1 into the main text to make our set of experiments on nine different datasets appear more prominently. We also reformulated the description of Figure 3 to make it more obvious that it presents a real dataset. We thank the Reviewer again for their comments.

---

### Official Review · AnonReviewer3 · 2020-10-29
**The paper does not offer a significant contribution**

**Rating:** 6
**Confidence:** 4

**Review:**

Summary: the authors study a number of neighbor embedding methods in terms of attraction-repulsion forces. The authors show that t-SNE, UMAP, FA2, and LE can be (approximately) unified as a common approach that use different levels of tradeoff between these two terms. They also discuss the increased attraction in UMAP as a result of negative sampling.


Review: a main portion of the technical contribution of the paper is simply writing down the gradients of the commonly used DR methods and providing an intuitive comparison of gradient terms. However, there is not much rigorous mathematical result to support the claim. I am aware of the complexity of such a strong theoretical result (e.g. the work of (Arora et al. 2018)). However, the current version of the paper does not offer any concrete result and remains on an intuition level observation. The experiments partially support the claims, but similar observations have been made before in related work. For instance, (Amid and Warmuth 2019) qualitatively sort DR methods by means of their "global score" (which is a notion of how well a method preserves cluster information).

A minor note on the experiments: in order to eliminate the bias in initializing some methods with PCA, I encourage the authors to also consider random (shared) initialization for all methods (Kobak and Linderman 2019). Also, since the paper is mainly experimental, (larger scale) experiments on a more diverse set of datasets would improve the paper.

---

> ### Author Response · Authors · 2020-11-13
> **Author response**
>
> We thank the reviewer for their review. Conceptually we disagree on three counts:
>
> 1) We strongly disagree that our observations are only intuition-level and not supported by theory. For example, we present a mathematical analysis of UMAP optimization, prove that it effectively decreases the repulsion, and then re-implement UMAP using Barnes-Hut to demonstrate this experimentally.
>
> 2) Of course whether a contribution is “significant” enough is in the eye of the beholder, but we would like to draw reviewer’s attention to the fact that the entire manifold learning community + the entire transcriptomic community is convinced that UMAP separates clusters and highlights continuous structures better than t-SNE due to mathematical properties of the cross-entropy loss. There are numerous articles and popular blog posts that make this claim. Our work shows that this is simply not the case; in fact, UMAP works *despite* the cross-entropy loss. This is not only novel, but also unexpected and important.
>
> 3) Amid & Warmuth 2019 is a very interesting work (which we cite) and the “global score” that measures similarity of the embedding to the first two PCs is useful. However, in their experiments, t-SNE and UMAP get around the same “global score”, so we don’t see how this is similar to our claim that t-SNE and UMAP have markedly different (orders of magnitude!) attraction strength.
>
>   Furthermore,  it would be interesting to approximately position TriMap (Amid & Warmuth 2019) on the attraction-repulsion spectrum, and inspection of Figure 5 of Amid & Warmuth 2019 suggests that TriMap has stronger attraction than UMAP (e.g. for Fashion MNIST the TriMap embedding looks remarkably close to the FA2 embedding shown in our paper). We would be happy to add some further discussion of TriMap into our Discussion.
>
> Regarding the experiments, we would like to point out two things:
>
> 4) The reviewer seems to have overlooked that we *already* did the control experiment they are suggesting, i.e. we already used shared random initialization for our Figure 2. All our conclusions stay the same, irrespective of initialization. We can point that out more clearly in the text.
>
> 5) The reviewer may also have overlooked all the other datasets shown in the Appendix. In total we analyze nine different datasets, and it is not entirely clear to us what a “more diverse set of datasets” would be. The results, as summarized in Table S1, are strongly consistent across datasets. We are happy to include additional datasets into our comparison if the reviewer has specific suggestions.

---

> > ### Comment · AnonReviewer3 · 2020-11-24
> > **Thank you for your response**
> >
> > Thank you for pointing out the random initialization experiments. I agree with Reviewer4 that the trade-off between attraction-repulsion is a known argument (for instance, see Venna at al. 2010). Your analysis of the form of the gradient (in my humble opinion) does not qualify as "theoretical analysis". On the experiment side, we should consider whether your analysis would offer any advantage to a practitioner to obtain more accurate visualizations. Again, your paper doesn't offer much in that regard, other than showing in some cases t-SNE and UMAP results might be over exaggerated (again, a known fact). However, I acknowledge that you have put tremendous effort for conducting these experiments and I would like to raise my score to 6 to encourage the community in making an effort to understand DR methods more extensively.
> >
> > Venna at al. Information Retrieval Perspective to Nonlinear Dimensionality Reduction for Data Visualization, Journal of Machine Learning Research, 2010.

---

> ### Author Response · Authors · 2020-11-24
> **Revision uploaded**
>
> We have uploaded a revision, moving Table S1 into the main text to make our set of experiments on nine different datasets appear more prominently. We have also inserted some comments  about the TriMap algorithm (Amid & Warmuth 2019) into the Discussion, speculating about how TriMap might fit into our framework. We hope that the confusion regarding random initialization have been resolved in our previous reply. We thank the Reviewer again for their comments.

---

### Decision · Program_Chairs · 2021-01-07
**Final Decision**

**Decision:**

Reject

**Comment:**


This paper analyzes several neighbor embedding methods-- t-SNE, UMAP, and ForceAtlas2-- by considering their objectives as consisting of attractive and repulsive terms. The main hypothesis is that stronger repulsive terms contribute towards learning discrete structures, while stronger attractive terms contribute towards learning continuous/manifold structures. The paper empirically explored the space parameterized by the relative weighting of the attractive and repulsive terms for the t-SNE and UMAP algorithms, using several data sets, and qualitatively confirmed their conclusions about the impact of the attractive and repulsion terms as the relative weights vary.

The experimental validation of the paper's main hypothesis is thorough and the use of diverse data sets and neighbor embedding methods is appreciated-- as the authors point out, several reviewers missed this contribution. However, several reviewers point out that the insight presented in the paper is already largely present in the literature, and that beyond its analysis the paper does not present new algorithms based on this insight. The authors rebut this claim by arguing that the novelty of the paper lies in it: (1) showing the contrary to the established opinion, UMAP works despite, instead of because, it uses cross-entropy loss, and (2) the paper offers for the first time a theoretical understanding of why ForceAtlas2 highlights continuous developmental trajectories, and (3) prior work has not made the connection between UMAP, ForceAtlas2, and t-SNE or suggested using exaggeration throughout the optimization process for t-SNE rather than simply as a warm-up. The paper does indeed present intuitions for (1)-(3) based on the attraction-repulsion ideas, and makes the connection between these neighbor embedding algorithms by viewing them as variations on the theme of attraction-repulsion, but these intuitions are not significant steps forward with respect to what is already known about how neighbor embeddings balance attraction and repulsion. The mathematical analyses consist of stating the gradient for the algorithms and explaining how weighing the attraction and repulsion terms differently lead to different qualitative observations. The use of exaggeration throughout the optimization process is straightforward, and no strong mathematical characterization of the properties of the resulting algorithm is given.

It is recommended that this paper be rejected, as it consists of a thorough empirical validation of an understanding of the trade-off between attractive and repulsive forces in neighbor embedding methods that was already present in the literature, along with some straightforward arguments connecting several popular neighbor embedding methods, but does not introduce any significantly new actionable insights or novel algorithms.